# UV-Curable Bio-Based Polymers Derived from Industrial Pulp and Paper Processes

**DOI:** 10.3390/polym13091530

**Published:** 2021-05-10

**Authors:** Lorenzo Pezzana, Eva Malmström, Mats Johansson, Marco Sangermano

**Affiliations:** 1Department of Applied Science and Technology, Politecnico di Torino, Corso Duca degli Abruzzi 24, 10129 Turin, Italy; lorenzo.pezzana@polito.it; 2Department of Fibre and Polymer Technology, KTH Royal Institute of Technology, School of Engineering Sciences in Chemistry, Biotechnology and Health, Teknikringen 56–58, SE-100 44 Stockholm, Sweden; mavem@kth.se (E.M.); matskg@kth.se (M.J.); 3Wallenberg Wood Science Center, Teknikringen 56-58, SE-100 44 Stockholm, Sweden

**Keywords:** pulp and paper industry, lignin, UV-curing, 3D printing

## Abstract

Bio-based monomers represent the future market for polymer chemistry, since the political economics of different states promote green ventures toward more sustainable materials and processes. Industrial pulp and paper processing represent a large market that could advance the use of by-products to avoid waste production and reduce pollution. Lignin represents the most available side product that can be used to produce a bio-based monomer. This review is concentrated on the possibility of using bio-based monomer derivates from pulp and the paper industry for UV-curing processing. UV-curing represents the new frontier for thermoset production, allowing a fast reaction cure, less energy demand, and the elimination of solvent. The growing demand for new monomers increases research in the environmental field to substitute for petroleum-based products. This review provides an overview of the main monomers and relative families of compounds derived from industrial processes that are suitable for UV-curing. Particular focus is given to the developments reached in the last few years concerning lignin, rosin and terpenes and the related possible applications of these in UV-curing chemistry.

## 1. Introduction

Petroleum-based monomers are currently the most available source of both thermoplastic and thermoset plastic materials. The growing awareness of pollution and environmental problems arising from the use of petroleum-derived materials is driving scientific research toward the replacement of commonly used polluting products with bio-based ones. The US department of Agricultural and Energy has set a target to increase the percentage of bio-based materials as sources of chemicals and materials by as much as 25% by 2030 [1].

The exploitation of the biomass of plants can represent an attractive source of bioenergy as well as bio-based chemical precursors [2]. This is particularly important considering that plant biomass is the most abundant renewable feedstock on Earth. For this reason, the pulp and paper manufacturing industry is being considered as a source for new vitalizing markets [3].

The reason for choosing the pulp and paper industry sector is connected to the large availability of bio-sources, which make it possible to forecast new polymer production products. Figure 1 shows the main constituents of waste liquor, the by-product of paper and pulp industry [4]. The relative amounts and characteristics of the main fractions are related to the process used to produce the paper [5,6,7].

Lignin is the second most abundant polymer in the world, and around 50 million tons of lignin are provided by the pulp and paper industry, of which less than 2% is used to make chemical value products [8,9]. The other 98% is used for energy generation purposes. Lignin has conventionally been considered as a low-value waste product, but it may be proposed, in this new scenario, as an interesting polymeric precursor. In fact, lignin is a complex aromatic polymer that contains different reactive groups, such as phenylpropanoid entities and carbon–carbon bonds, which can be exploited as polymerizable reactive groups [3].

Looking at the by-products of the pulp and paper industry, there is another important family of compounds, called rosins, which may be interesting as a source of bio-based polymer precursors. Rosin is an important candidate for obtaining polymerizable structures for both linear and crosslinked materials [9]. More than one million tons of rosin are produced annually. Rosin is mainly used as a constituent of inks, varnishes, adhesives, paper size, cosmetics and medicines, but rosin itself and its derivatives may also be proposed as reactive monomers for the production of polymers [10].

The volatile fraction of the resin, turpentine, which is composed of a mixture of terpenes, should also be considered as a valuable organic feedstock. Turpentine is by far the most frequently used source of terpenes, whose yearly production throughout the world amounts to some 350,000 tons. Certain derivatives, such as α-pinene, β-pinene, limonene and myrcene, have been studied as starting materials for the synthesis of polymers, and α-pinene (45–97%) and β-pinene (0.5–28%) are the main products [11]. Pinenes and limonenes are cheap and abundant natural products, and are used extensively as chemical precursors for a wide variety of polymerizable monomers [9].

The above cited family of compounds is becoming more attractive as a substitute for petroleum-derived monomers. A huge amount of work has already been done to exploit their use in several different application fields, ranging from linear polymers, such as polyester, polyurethane and polycarbonates, to crosslinked resins, such as vinyl ester, epoxy and acrylate resins [12,13,14,15,16,17].

This review is focused on the following three main families of products obtained from industrial pulp and paper processing: lignin, rosin and terpenes, as they can be exploited as photocurable starting materials.

The use of the UV-curing technique, instead of the thermal-curing method, is particularly interesting because of its reduced energy consumption, high cure speed, even at room temperature, and the absence of VOC emissions [18]. The following sections deal with the significant families of compounds that can be processed from the paper and pulp industry in order to create a natural base for UV-curable materials.

### 1.1. Lignin and Derivatives

Cellulose, hemicellulose and lignin made up the lignocellulosic biomass. Lignin accounts for about 15 to 30% of the wood constituents, as shown in Table 1. Its structure changes according to the biomass family. Softwoods are richer in lignin, whereas hardwoods are richer in hemicellulose [19]. Cinnamic alcohols are the main constituents of lignin; this family is composed of p-coumaryl alcohol, coniferyl alcohol and sinapyl alcohol [1]. Grasses contain all three components, while softwood mainly has coniferyl alcohol, and hardwood contains both coniferyl and sinapyl alcohol. Table 2 shows the different percentages of monolignol present in the three main wood families, that is, broadleaf wood, conifer wood, and grasses [20].

In addition to the principal phenolic nuclei, considerable amounts of other components can also be found in lignin, such as coniferaldehyde, sinapaldehyde, p-hydroxybenzoate, ferulate, p-coumarate, hydroxycinnamates, and other by-products from incomplete monolignol biosynthesis [20].

The most common linkage, the β-O-4 ether linkage, which typically accounts for 50% of the bonds, is created during the polymerization reaction, and it is the typical target of a degradation process. Figure 2 shows the principal linkages between the different aromatic units present in lignin [1,8].

Lignin can be separated by chemical, physical and biological processes [21]. The strategy used to reduce lignin has a great impact on the chemical structure of the resulting products. Figure 3 shows a scheme of the catalytic processes that could be conducted to obtain chemicals and fuels. Several industrial sources of lignin, which have different features according to the used method, are available [2,3,22,23]. The main ones are:Kraft lignin (KL), which is a by-product of the paper and pulp industry after lignocellulose has been treated at an elevated temperature (170 °C) and pressure. KL is becoming the added value product of this industry, transforming the entire process into a bio-refinery system.Lignosulfonates (LS), which are derived from the sulfite process, are obtained through a neutral or acidic treatment/cooking. Sulfite pulping is the second most commercially-used process, and it produces about 7 million tons of lignin annually.Soda lignin (SL), which is generated as a co-product from flax, straw, and/or non-wood fibers, using anthraquinone as a catalyst. However, SL has a low production capacity, due to the annual variability of the feedstock.Organosolv lignin (OL), which is obtained from pulp by means of an organic solvent treatment. In this process, the lignin is obtained using solvents, using neither acidic or alkaline conditions, and it is an alternative process to the pulping technology. The extraction of lignin with an organic solvent mainly involves breaking down the α-aryl ether bonds. OL is usually less contaminated than other types of technical lignins, and with recovery of the solvent it is a closed process.Steam-explosion lignin (SEL), which is derived from a steam process. The process involves the impregnation of lignin with steam at high temperature and pressure. This leads to an increase in the reactivity of the substrate, allowing more enzyme access and digestibility in order to separate the components.

The previous sections consider methods used in the chemical degradation of lignin to yield a soluble and isolable product from a biomass, which is generally known as technical lignin. This section instead focuses on the degradation methods and chemical processes necessary to convert technical isolated lignin into useful (chemical) products. Several catalytic strategies are aimed at obtaining high yields and the selective production of defined aromatic monomers from lignin and lignocellulose. The most important strategies are:Hydrogenolysis, which is a pyrolysis process that results in the formation of small fragments in the presence of hydrogen. Cleavage refers to the C–C bond [17]. Typical temperatures of around 300–600 °C are used. Catalysts, such as Ru/C, Pd/C or Pt/C, are used to improve the final yield of the product, and bimetallic systems have also been explored. An appropriate choice of the catalyst leads to a high degree of depolymerization and a high product yield. The hydrogenolysis of the C–C bond is catalyst dependent, as shown in Figure 4 [8].Oxidation, whereby lignin is generally oxidized with nitrobenzene, hydrogen peroxide or a metal oxide. Some methods use heterogeneous catalysts, such as metal or surface-supported metal, Pt, Pd, Re, Ti, Ni and Cu. Homogeneous catalysts include meta and organic compounds; ionic liquid has also been used as an alternative catalytic system [1].

Only three products are produced commercially on a large scale from lignin: vanillin, dimethyl sulfide and dimethyl sulfoxide [2]. Of these, vanillin is currently one of the only molecular phenolic compounds manufactured from biomass on an industrial scale, and it has the potential of being a key renewable aromatic building block [24,25]. The oxidative cleavage of lignin to produce vanillin is one of the oldest known depolymerization processes [8]. Relevant vanillin derivatives can be isolated under pulping conditions, while vanillic acid can be isolated under a strongly oxidizing condition. Vanillin alcohol could instead be obtained under reducing conditions [1].

Most catalytic methods have focused on facing the fundamental challenge of selective bond cleavage in organosolv lignin or lignocellulose, focusing on the β-O-4 moiety. However, more research is required for the development of robust and recyclable catalysts. Novel catalytic systems should be studied to fully valorize all the lignocellulose components. Techno-economic analyses should be performed for these processes to understand the utility and possible new products that may be derived from the emerging building blocks.

Aromatic compounds are key intermediates in the manufacturing of polymers, and the synthesis of these derivatives from lignin is therefore one of the major topics of interest in this field, since lignin is the main source of aromatic bio-based substrates. Phenols can be obtained with a variety of chemical structures from lignin deconstruction, as shown in Table 3; besides vanillin, as previously mentioned, ferulic acid (FA) is the main one [26]. Other products, such as sinapic acid and its derivatives, can be isolated from lignin depolymerization mixtures [8]. A Fenton modification improves the yields of phenolic monomers, especially for ethyl-p-coumarate and ethyl-ferulate [21].

Ferulic acid is a component of lignin, and it is part of the hydroxycinnamic family, together with sinapic and caffeic acid. It is a key component of the cell wall and it has several useful functions for the life of a cell, such as antioxidant and free radical scavenger properties [27,28,29,30]. New methods are being implemented to isolate these components from sugarcane bagasse, bark trees and kraft lignin obtained from the pulp and paper industries. These methods can involve chemical and physical approaches, such as hydrolysis and extraction, or bacterial action [31,32,33,34,35]. An alternative method could be that of resorting to synthetic biology to valorize lignin and biosynthesized coumarins [36]. Another bio-source of FA could be agro-industrial waste, which represents a large, cheap and available source of chemicals [29,37].

### 1.2. Rosins and Its Derivatives

Rosin is a component of conifer tree resin and it is also known as colophony [38]. Pine trees (*Pinus* genus), which are widespread in the Northern Hemisphere and are adopted intensively in the pulp industry, are the most important source of rosin [39]. The main structures of the different derivates are built on abietane and pimarane skeletons, as shown in Figure 5 [38,40].

Gum rosin and tall oil rosin are the two most industrially important types of rosin. Gum rosin is the non-volatile fraction of conifer resin [11]. Tall oil rosin is a by-product of wood pulping in the kraft process, and it accounts for 35% of the total production of rosin, which is around 1.2 million tons per year. Pine trees are used extensively in the pulp industry. Wood rosin is the least available type of rosin. It is extracted from mature pine stumps, which are chipped and soaked in a solvent, after which solid wood rosin is obtained by means of a distillation process [10].

### 1.3. Terpenes

Terpenes are mainly synthesized by plants, but also by a limited number of insects, marine micro-organisms and fungi. The chemical composition of turpentine depends to a great extent on the tree species, geographic location and the overall procedure used to isolate it. The major components are a few unsaturated hydrocarbon monoterpenes (C_10_H_16_), namely, α-pinene (45–97%) and β-pinene (0.5–28%), and smaller amounts of other monoterpenes [41]. Most terpenes have a cycloaliphatic structure with isoprene as the elementary unit, as shown in Figure 6. Terpenoids can be considered as modified terpenes [11]. Only a few terpenes are at present under analysis to produce bio-based polymers, despite their low cost, abundance and large variety [10].

Recent studies have demonstrated that it is possible to isolate different products, in particular using the waste of the sugi wood-drying process (BWP), thereby making BWP a source of these useful compounds. Sugi is the one of the most common conifers in Japan [42,43].

## 2. UV-Curable Monomers from the Pulp and Paper Industry

Among the different possibilities of using lignin, rosin, terpenes and the relative bio-based derivates obtained from the pulp and paper industrial process, this review has focused on UV-curable applications. Several works illustrate the role of such applications in polyesters, polyurethanes, epoxy resins, foam, hydrogels and aerogels [1,3,22,23]. However, there is a lack of information about this specific approach to obtain different bio-based products.

Therefore, the next sections are focused on new approaches to UV-curing. In particular, the first section is on the use of lignin as a raw material. The second section is on vanillin, since this bio-based monomer is already made at an industrial scale and is very attractive as a UV-curing resin. The third section is devoted to rosin, and the final one to terpenes.

### 2.1. The Photopolymerization of Lignin and Its Derivatives

The use of lignin has developed over the few past decades. Several groups have started to exploit biomass as an alternative source to petroleum-based materials. In 1989, Hatakeyama et al. [44] presented the relationship between the chemical structures and the physical properties of the new types of high-performing polymers that had been synthesized from degraded lignin products. The complex nature and the heterogeneity of the reagent were the reasons for the difficulties of using this biomaterial as a source to produce polymers. In 1998, Gandini et al. discussed the synthesis of polyesters and polyurethanes based on lignin of various origins [45]. After the first studies, the same group investigated different kinds of biomass as resources for various synthetized polymers. They employed organosolv lignin in ink, paints and varnishes in order to study the effect on these systems. The main results were a linear increase in viscosity and a reduction of misting as a consequence of adding lignin to the composition [46,47]. In 2006, coatings and composites were formed using lignin derived from sugarcane bagasse. Thermal curing was employed to form the different films [48]. The same group investigated the substitution of phenol in phenol formaldehyde resin with lignin. The acetylated lignin obtained from sugarcane bagasse showed an increase in the water resistance of the formed coatings [49]. Cazacu et al. [50] and Laurichesse et al. [51] described the employment of lignin in the chemical and polymer industries, highlighting the advantages and disadvantages of using biomass instead of petroleum-based sources. Most of the cited works reported the use of lignin while exploiting the thermal curing method; it is only in recent years that lignin has starting to be used in UV applications.

Alsuami et al. wrote a review on the use of lignin in the UV-curing technique [52]. The same group developed a study on the reactivity of lignin in photopolymerization. Lignin can also be used as a filler in UV applications. In such a case, no chemical functionalization is performed, and lignin, which is used in the extracted form without modification, acts as a reinforcement. Rozman et al. [53] studied the utilization of lignin as a filler in UV-curable systems in the presence of both a cationic photoinitiator and a free radical photoinitiator. The stiffness and resistance to abrasion improved as the lignin content in the photocurable formulation increased, due to the intrinsic stiffness of lignin and the presence of phenol groups. Zhang et al. [54] incorporated lignin in methacrylic resin at various concentrations (1 wt % as a maximum). The positive effect on the mechanical properties was related to the ultimate strength, which increased by 46–64%, and to an enhancement of Young’s modulus of 13–37%. Fracturing the samples revealed that a certain amount of lignin was favorable for dissipating the stress concentration. This could be one of the reasons for the improvement in the mechanical performance of 3D-printed composites which include lignin. Ibrahim et al. [55] exploited organosolv lignin in polyurethane resins. The important outcome was the enhancement of the mechanical properties of the resins with lignin. The optimal concentration was 0.6%.

Chao et al. synthesized lignin-based waterborne polyurethanes [56]. A positive effect of the lignin was found in the mechanical properties, and its addition influenced the gloss and the light transmittance of the cured films.

Wang et al. [57] prepared UV-curable lignin thermoplastic copolymers, as shown in Figure 7. A benzophenone acrylate derivative, ABP, was selected as a photosensitive comonomer. In a general photo-cross-linking mechanism, ABP is excited by the absorption of a photon, and the excited benzophenone abstracts aliphatic hydrogens from neighboring polymer chains, and thus generates radicals. The photogenerated radicals combine to produce a cross-linked network. The UV-cured films were 100 μm thick; under UV irradiation, up to ~30 wt % of the lignin-g-poly(styrene-*co*-ABP) copolymers cross-linked to form polymer networks. These results are expected to provide a convenient and robust alternative to lignin-based functional coatings, with enhanced surface hardness, solvent resistance and thermal stability.

Hajirahimkhan et al. [58] optimized the synthesis of methacrylated lignin through a central composite design and response surface methodology. The chosen lignin was kraft (KL) from a pulp waste process. The obtained product was used in a UV-curable system. Methacrylic lignin (30 wt %) was added to a UV-curable coating formulation, containing a hexafunctional methacrylated siloxane-based crosslinker (EB-1360). The presence of lignin induced an increase in hydrophobicity, an improvement of the double bond conversion (reaching 64%), and better thermal stability.

Hajirahimkhan et al. [59], in a subsequent work, developed another lignin-based UV-cured coating. The selected type of lignin was, as before, KL, which was methacrylated with methacrylic anhydride, as previously reported [58]. They successfully prepared UV-cured coatings using up to 31 wt % methacrylated lignin in the formulations (Figure 8). The addition of lignin improved hydrophobicity, induced higher thermal stability/char formation, and enhanced surface adhesion of the UV-cured coatings.

Yan et al. also studied the use of lignin in coating applications [60]. They used organosolv lignin to prepare UV-curable coatings. They successfully synthesized lignin-based epoxy acrylate oligomer (LBEA) in two steps. First, lignin reacted with bisphenol A diglycidyl ether (DGEBA) to produce a lignin-based epoxy (LBE). In the second step, LBE reacted with acrylic acid to produce LBEA. The insertion of lignin improved the pencil hardness, as well as the flexibility, adhesion and chemical resistance of UV-cured films. Hence, lignin was confirmed to be a favorable new biomaterial that could be exploited in UV-curable formulations.

Yan et al. [61] used organosolv lignin (OL) to synthesize organosolv-lignin-based epoxy acrylate (OLBEA) coatings. OLBEA was prepared, as before, in two steps. In their study, the curing process was a UV-thermal dual-cured strategy: the UV curing was performed first, and then a final thermal post-curing process was done on the crosslinked films. The OL content in the formulations was varied between 5 and 25 wt % and mixed with bisphenol-A-diglycidyl ether diacrylate (BADGE). The mechanical properties of the photocured films improved as the modified lignin content in the photocurable formulation increased; the hardness, adhesion and flexibility of films were enhanced.

Some researchers have studied the exploitation of functionalized lignin for 3D printing technology. Sutton et al. [62] used organosolv lignin from pulp-grade wood chips to generate new photoactive acrylate resins. The resins were a mixture of commercially available resin components with up to 15 wt % of lignin. The lignin was functionalized, by means of methacrylation, using methacrylic anhydride. The difference in color, when the lignin-based resin was varied, is shown in Figure 9.

Hence, methacrylic lignin resin is compatible with 3D printing, and it could have a great potential as a binding agent to improve printing quality. A challenge for future work is the correlation between penetration depth and critical cure dosage, which depends on the amount and chemical structure of the lignin in each sample.

All these works, which are listed in Table 4, showed good results when using lignin as a co-monomer in photocurable formulations. Indeed, the addition of modified lignin enhanced the final properties of the new UV-cured materials, at least for some of the features, such as hydrophobicity, crosslink density and thermal stability. Thus, lignin can be considered a promising UV-curable bio-based material, and it is worth doing further and deeper investigations for future green developments.

As mentioned before, vanillin is one of the monomers available from lignin for industrial purposes, and it has been investigated by means of a UV-curing technique [26].

Ding et al. [63] studied the behavior of a sustainable thermosetting resin of natural phenolic (meth)acrylates derived from softwood lignin. The bio-derived monomers were: guaiacol, vanillin alcohol and eugenol. Acrylation was performed on guaiacol (G) and vanillin alcohol (V) with methacrylic anhydride, see Figure 10. On the other hand, thiol-ene click chemistry was performed on eugenol to exploit the allylic double bond (as shown in Figure 10), then the phenolic end groups were acrylated using acryloyl chloride. This allowed a novel acrylate monomer to be prepared from eugenol (E in Figure 10). These natural acrylates possess fast photo-curing rates, and the crosslinked material showed high thermal stability and good mechanical properties, thus ensuring a competitive alternative to commercial prototype resins for the stereolithography (SLA) 3D printing technique. The synthetic route is schematized in Figure 10, where the results of tests on binary and ternary formulations are also shown. The G and E monomers were mixed, in different molar ratios, to test the reactivity and the performance; a third component, a cross-linker, was then introduced to achieve the high photo-reactivity necessary for the SLA process. Two cross-linkers were selected, that is, a widely used commercial one (trimethylolpropane-trimethacrylate, T in Figure 10) and a natural synthesized one (vanillin-dimethacrylate, V in Figure 10).

The outcome of this work was that the natural phenolics functionalized from the biomass had excellent photocuring kinetics, which helped form the crosslinked networks that allowed the light-based 3D printing of mechanically robust and high bio-based content objects.

Bassett et al. synthetized a bio-based resin using vanillin-based monomers for SLA techniques [64]. They used methacrylated vanillin (MV) as a monomer, and glycerol dimethacrylate (GDM) as a cross-linker. The two molecules were mixed in a 1:1 mol ratio, and TPO was used as a photoinitiator. Photorheology was performed to optimize the printing parameters. Additionally, some samples were further thermally post-processed after the printing. The effect of post-processing after printing was evaluated, and an enhancement of the double bond conversion of 23% was observed. The vanillin-based resin, prepared via SLA with post-processing, showed the highest Tg (153 °C) and highest Young’s modulus (4900 MPa). It was also found that the post-processing induced an important enhancement of the final properties of a 3D-printed structure, thus demonstrating the potential for the use of resin as a robust material for the SLA process. Moreover, as a result of the low viscosity and low critical curing energy, the vanillin-based resin showed a potential for use as a component in the development of bio-based, high Tg, high strength materials for SLA.

Maturi et al. [65] developed a new resin for digital light processing (DLP)-based 3D printing. The important feature of this resin was the extremely high bio-based content of 96.5%. Itaconic acid, glycerol, 1,3 propendiol and vanillic acid were used as precursors in the photocurable formulation. Itaconic acid-based photocurable polyester, poly(1,3-propanediyl-co-glyceryl) itaconate-co-vanillate (PPGIV), was synthetized by reaction at a high temperature (145 °C) under a nitrogen atmosphere. The double bonds of itaconic acid represented the photocurable moieties; however, the concentration was too low to guarantee a feasible application in DLP. Thus, two different crosslinkers were synthesized, by means of the acrylation of itaconic acid and citric acid with 2-hydroxyethyl methacrylate (HEMA). The printed structure showed a better mechanical resistance and lower fragility than most of the commercial equivalents, thus demonstrating that there is no loss of performance when moving from fossil to bio-based sources. Moreover, a biocompatibility test was conducted and no cytotoxicity or sensitization stimulating immune cellular response was observed.

Navaruckien et al. [66] investigated a bio-based system with photocurable resins for optical 3D printing; therefore, novel vanillin acrylate-based resins were studied. Cross-linked polymers were prepared by means of the radical photopolymerization of vanillin dimethacrylate, VDM, and vanillin diacrylate, VDA. The photo crosslinking of the VDM resins was faster than with the VDA resins. The vanillin diacrylate-based resins showed a higher double bond conversion, and the crosslinked materials showed high thermal stability and better mechanical properties than the vanillin dimethacrylate-based photocured films. Moreover, the vanillin diacrylate polymer film showed a significant antimicrobial effect. Two types of optical printing techniques were used to produce 3D objects out of custom-made photo cross-linkable resins: direct laser writing (DLW) and the microtransfer molding technique (μTM), which is also known as nanoimprint lithography, see Figure 11.

Ferulic acid is also used in several applications; it has antioxidant, anti-inflammatory, antimicrobial and antiallergic properties and it is thus widely employed for biological and pharmaceutical purposes [67,68]. Moreover, FA can be used as a base for different chemicals, from polyester to epoxy [28,69,70,71,72]. Another important feature of FA and its derivates is the presence of a cinnamoyl double bond. This allows ferulic derivates to be exploited as monomers for UV-curable formulations or as reactive moieties, since they can go under [2 + 2] cycloaddition. The [2 + 2] cycloaddition is a dimerization that takes place when the molecule is exposed to UV: the C=C bond breaks and creates a new cyclobutane ring that links two molecules. The cinnamic derivatives have also been used to produce polyesters, polyamides and poly(anhydride esters), and many other types of polymers [69,73].

Teramoto et al. [74] synthetized a trehalose cinnamoyl ester by exposing the monomer to UV. The dimerization was confirmed by means of a UV−Vis, FITR measurement. Moreover, an increase in the Tg revealed that photo-crosslinking had occurred. In a successive work, the same group, headed by Yano [75], photo-crosslinked cellulose using cinnamoyl moieties. This example of a possible cinnamoyl functionalization of bio-based monomers produced materials with high cell proliferation. New biocompatible and bio-based materials could be further explored starting from these studies.

Castillo et al. [76] synthetized novel polyamides, containing a cyclobutane ring, by [2 + 2] cycloaddition. After the functionalization of FA, the product was irradiated by stirring for 20 h under an argon atmosphere. Nuclear magnetic resonance spectroscopy (^1^H-NMR) and liquid ionization mass spectroscopy (LI-MS) analyses confirmed the presence of dimers. Tunc et al. [77] developed other photosensitive polyamides by using cinnamoyl moieties. The cross-linking reaction was reversible, and the functionalized aliphatic polyamides cross-linked photochemically at 364 nm could be de-cross-linked photochemically at 254 nm. This demonstrated the possibility of a reaction in two different directions being performed. The trigger was constituted in the different wavelength of the light, which allowed the properties to be modified and varied accordingly.

Hu et al. [78] developed a cyclic carbonate monomer containing cinnamoyl moieties. The copolymerization of the monomer was carried out with l-lactide. The kinetics was studied by means of UV–Vis and FTIR measurements, which confirmed the cycloaddition (the mechanism is presented schematically in Figure 12).

Nagata et al. [79] developed a photocurable biodegradable polyester The cast films were irradiated; the polymer network was formed since the cinnamoyl group underwent [2 + 2] cycloaddition, producing the cross-links. The gel contents increased as the photocuring time increased, reaching values of over 90% for all the films after 90 min of irradiation. These new films could be used for bio-medical or environmental applications.

Kim et al. [80] prepared a photocurable polymer by mixing a commercial epoxy resin and cinnamic derivates. Thin films were formed after exposing it to UV. The main features of the films were good thermal stability and optical transmittance.

Ding et al. [81] synthesized a trans-cinnamate epoxidized soybean oil (ESOCA) by reacting renewable biomass raw materials, trans-cinnamic acid and epoxy soybean oil. The decrease of the C=C peak was followed to confirm cycloaddition; the Tg of the film was 38.5 °C and good flexibility, adhesion and UV shielding performances were observed. This study demonstrated a new strategy to prepare flexible high-performance renewable biomass thermosetting materials.

Shibata et al. [82] synthetized allylated derivates of coumaric and caffeic acids. They used thiol-ene chemistry to achieve cross-linked networks.

Pezzana et al. [83] developed another bio-based coating using ferulic acid (FA). They used the allylation of FA in their study, and showed that thiol-ene chemistry allowed the resins to be cross-linked. They tested different bi- and tri-functional ferulic bio-based monomers. The photocured network was achieved by means of a reaction with a tri-thiol, as shown in Figure 13. The properties of the cured films were studied, and good adhesion and stability were observed. The highest Tg reached was 24 °C for the trifunctional ferulic derivate. This work demonstrated the feasibility of using ferulic acid as a starting green monomer to develop UV-curable coatings, with the possibility of modulating the final Tg of the crosslinked networks.

### 2.2. The Photopolymerization of Rosin

Rosin has been used for coating formulations for centuries. The unique properties of rosin are due to its hydrophobic skeleton and its hydrophilic carboxy groups. These features lead to its excellent solubility and compatibility with a variety of synthetic resins. Rosin derivates can be obtained by means of the appropriate modifications, and this renewable resource is therefore widely available to develop new resins for coatings applications [84].

Lee et al. studied one of the first examples of using rosin in UV-curable applications [85]. They synthesized functional monomers from gum rosin. They synthesized monofunctional acrylic rosin derivatives and a trifunctional acrylate, starting from maleopimaric acid anhydride (MPA) and fumaropimaric acid (FPA). Copolymers were also produced, by copolymerization between methyl methacrylate and monofunctional monomers. All the resins showed good solubility and absorbance in the UV region, and the photocuring method was therefore used to produce negative photoresists. After irradiation, the films were developed in a base solution. These results showed the possibility of functionalizing rosin-based monomers to produce different resins, with a large variety of properties and applications.

Do et al. [86] synthesized a photocurable hydrogenated epoxy methacrylate rosin (HREM). Glycidyl methacrylate was used to functionalize rosin via an epoxide ring opening reaction. The photopolymerization reactivity of HREM was studied by means of photo-DSC.

Do et al. [87], in a subsequent work, further explored the use of HREM to make a pressure sensitive adhesive (PSA). The rosin-based monomer was mixed, up to 30%, with a formulation of acrylic PSA. For the polymerizable photoinitiator, 2-(acryloyoxy) ethyl 4-(4-chlorobenzoyl) benzoate (P-36) was used. The Tg of HREM and the modulus of the PSA/HREM blends were sufficiently low, and the PSA molecular mobility and its crosslinking efficiency increased after blending with HREM. The PSA/ HREM blends maintained their Tg in the sub-zero temperature range, with high wettability, and the properties and probe tack of the PSAs increased remarkably as the HREM content increased, compared with the PSA/hydrogenated rosin blends.

Another group, led by Ahn [88], investigated the effect of rosin on adhesive formulations. Cationic polymerization was exploited, and the bio content was extremely high, i.e., 97%. The peel and tack properties were similar to those of commercial tapes, and the formed tapes had a significantly stronger shear strength than commercial tapes.

Liu et at. [89] developed a polyurethane acrylate system with hydrogenated rosin. The Tg was 54.4 °C, and the positive effect of the rosin resulted in less shrinkage than that of a commercial oligomer, and a better adhesion to glass. Moreover, the formulation was highly UV reactive, thus indicating a promising application for this type of oligomer.

Lu et al. [90] synthesized a rosin allyl ester in order to produce a bio-based resin for UV curing. The cured film showed good properties (Tg of 24 °C), with a thermal decomposition that started at 264 °C. This could therefore be a good example of a soft monomer with superior properties, in terms of flexibility, impact strength and adhesion. Lu et al. [91] also used a new method to functionalize rosin derivates. Microwave irradiation led to the production of an allyl derivate of levopimaric acid. Monofunctional and bifunctional monomers were produced. Certain synthesis parameters, such as microwave power, temperature, reaction time and catalyst dosage, were optimized. The UV-cured products (see Figure 14) were formed by mixing synthesized allyl acrylpimarate with 7% Michlers ketone (MK). Other photoinitiators were also tested, but the results were poor. Rosin molecules can improve chemical resistance, adhesion, and certain mechanical properties, such as thermal stability and hardness. Indeed, the thermal decomposition started at 292 °C for the crosslinked resins and at 263 °C for the linear polymer, while the Tg of the cured products were 49 °C and 11 °C, respectively.

The same group developed another rosin-based resin for UV curing [92] (see Figure 15). In this case, mono and tri-allyl maleopimarates were synthetized. The introduction of the rosin structure into polymeric UV-cured films improved the adhesion and mechanical properties. The tri-allyl maleopimarate polymers showed good chemical stability and a great potential for coating applications.

In 2019, Lu et al. [93] provided a route to produce a new bio-renewable resin. Isopimaric acid, methyl isopimarate and allyl isopimarate were synthesized and used for UV-curing. They prepared the sopimaric acid by means of selective crystallization. They synthesized the methyl isopimarate and allyl isopimarate using isopimaric acid as the raw material. The monomers were mixed with the photoinitiator in a fixed proportion using tetrahydrofuran as the diluent. The glass transition temperature of the UV-cured product derived from methyl isopimarate was lower than that of the polymer of isopimaric acid, and since the carboxylic acid group was converted into an ester group, it became more flexible. On the other hand, the UV-cured product derived from allyl isopimarate was generated from bifunctional monomers, which provided a comparable glass transition temperature with that of the product from isopimaric acid. The important result of this work was the formation of new types of natural rosin monomers, which showed high potential for the substitution of petroleum-based polymer products.

Lu et al. [94] developed a two-step 3D-printing approach, to prepare a thermoset derived from cellulose and rosin, by means of UV-induced chain-growth polymerization and step-growth polymerization. They used the rosin to create a novel acrylate monomer named dehydroabietic acid glycidyl methacrylated (DAGMA). DAGMA was tested as a monomer with a percentage of 2-hydroxyethyl acrylate (HEA), while methacrylated cellulose (CMA) was used as a crosslinker in chain-growth polymerization and hexamethylene diisocyanate (HDI) was applied as a cross-linker in step-growth polymerization to form a dual-cure network in 3D-printed thermosets, as shown in Figure 16. Different HEA/DAGMA ratios were analyzed, and the rosin content was proportional to the increase in the mechanical strength and toughness of the 3D-printed thermosets. A mechanical test showed that the dual-cure network could lead to phase separation and greatly increase the mechanical and thermal properties of 3D-printed thermosets. 3D-printed thermosets exhibited excellent shape memory and repairability, and the repair efficiency of the mechanical strength was up to 95.2%. A unique characteristic, originating from the rosin moiety, was a strong luminescence from aggregation-induced emissions (AIE). As a final experiment, hydrogels were made through degradation of the thermoset material. The chemical degradation of cellulose, using a NaOH solution, led to hydrogels. These 3D-printed thermoset-derived hydrogels have shown a great application potential for flexible electronic and smart photoelectric materials.

### 2.3. The Photopolymerization of Terpenes

In the last few decades, a great deal of work has been conducted by James Crivello’s group to investigate the reactivity of epoxidized limonene and other terpenes in cationic photopolymerization using iodonium and sulfonium salt.

In 1995, Crivello et al. [95] investigated the reactivity of monoterpenes. They converted α-terpinene, γ-terpinene and limonene to their corresponding diepoxies and photocured them by cationic polymerization. They found different behaviors for the three components of the same family; the derivates from γ-terpinene and limonene underwent ring-opening polymerization of both epoxy groups to give crosslinked polyethers as the predominant process while the epoxide from α-terpinene underwent an intramolecular rearrangement to give the epoxy ketone that did not polymerize. Similar behavior was predicted for other 1,3-monoterpene diepoxides.

In a successive work [96], they used α-terpineol as a reagent to produce allyl and propenyl derivates. Cationic photopolymerization was performed with the synthesized monomers, and a good reactivity was revealed.

Crivello et al. also studied other terpenes [97]. They synthesized allyl, allyl epoxide and propenyl epoxide from nopol. Nopol is an optically active bicyclic primary alcohol, used in soap fragrances as well as in agrochemical industries for the synthesis of pesticides and household products. Nopol is a bio-renewable, inexpensive and easily available substrate that can provide a suitable reagent for functionalization. Crivello demonstrated the high reactivity of epoxy, allyl and propenyl monomers toward cationic polymerization. This work paved the way toward further studies and research in UV-curable networks.

An epoxy alcohol was subsequently used to form hyperbranched polymers. Naturally occurring terpene alcohols were epoxidized and they showed a marked reactivity toward cationic photopolymerization. This study showed the possibility of increasing the reactivity by having an alcohol function in an epoxy cyclohexane ring system [98].

Park et al. [99] investigated the reactivity of two epoxide derivates from terpenes, that is, α-pinene and limonene. The outcome of their investigation revealed a high reactivity to cationic photopolymerization initiated by iodonium salt. They found some side reactions, which limited the production of homopolymerization, although considering the two epoxides as comonomers could confer benefits, in term of viscosity, polymerization rate and induction period. Moreover, they could change the mechanical properties of the epoxy-based photopolymerized crosslinked network.

Tehfe et al. [100] developed a free-radical-promoting cationic polymerization process for two epoxy monomers, one of which was limonene dioxide. The terpenes reacted with epoxidized soybean oil, by means of cationic polymerization, when iodonium salt was used. The novelty of this work lay in the possibility of operating under sunlight and of obtaining uncolored tack-free coatings. The same group considered the silyl radical chemistry, and developed a resin containing limonene oxide [101]. The results indicated a high reactivity to sunlight and selective irradiation over a narrower wavelength range. These results open the way toward very efficient epoxy systems. Lalevée et al. [102] investigated the radical chemistry of silyl in detail, and they overcame oxygen inhibition and developed renewable epoxy monomers for green chemistry applications.

Claudino et al. [103] investigated the potential use of limonene for thermosetting resin through thiol-ene chemistry. Limonene was used as a base for two different prepolymers, as shown in Figure 17. The intrinsic difference in reactivity of the two unsaturations in limonene was considered an advantage to form a branched oligomeric thermoset precursor. The choice of stoichiometry and the thiol functionality allowed different resin structures to be formed. The second step was the crosslinking of the macromonomers via UV. The authors showed the possibility of tuning the final properties according to the initial stoichiometry between the macromonomers and the choice of the thiol crosslinker.

Breloy et al. [104] made a green coating using a limonene derivative. They employed cationic polymerization and thiol-ene chemistry to achieve the formation of a film, (Figure 18). They used both mono- and di-epoxide limonenes. Bis(4-methylphenyl)iodonium hexafluorophosphate was used as the cationic photoinitiator. The final conversions obtained for both the epoxy and allyl groups were extremely high (>70%). The irradiation time required for tack free coatings was 20 min. A final study with eugenol was carried out to prove the antibacterial effect of this natural monomer. The incorporation of eugenol led to a marked decrease in bacterial adhesion.

Stamm et al. [105] functionalized a-pienene to generate a renewable material. They used the enzymatic route to synthesize sobrerol. Methacrylation was then conducted and the sobrerol was finally polymerized into poly(sobreryl methacrylate) (PSobMA) (Figure 19), using different radical polymerization techniques. The PSobMAs were mixed with a trifuncional thiol and a radical initiator (Irgacure 651) to assess the possibility of utilizing the ene bond in the side-chain for crosslinking purposes. Successive tests, FT-Raman spectroscopy and solubility investigations confirmed the curing reaction.

Weems et al. [106] used vat polymerization to produce 3D printed structures with terpene and terpenoid. They used limonene, linalool, nerol and geraniol as base monomers for the resin. Prepolymers were formed by reacting half of the thiol groups with the same quantity of alkenes. The remaining half of the thiol resin was added to the prepolymer to make a resin. The crosslinking time for limonene and linalool-based resins was found to be approximately 5 s, while the nerol- and geraniol-based resins crosslinked over the course of 1 h, under 3D printing conditions. Thus, the linalool-based resin was the most effective for printing, due the combination of its low viscosity and its reactivity, with limonene requiring longer exposure times to induce photo-crosslinking. Figure 20 reports the objects printed for the different resins. The prepolymer technique was used to print limonene-based material. Nerol, geraniol, and the prepolymers were not suitable for printing into 3D structures because of the slow reaction kinetics under 3D printing conditions.

Weems et al. [107] later also studied the use of myrcene for 3D printing. In the same way as in the previous work, myrcene also suffered from low viscosity and low reactivity. Hence, it was polymerized to develop a polymyrcene in linear and branched form that was suitable for printing. They adopted free radical and anionic polymerization. They tuned the properties by changing the thiol to cross-link or functionalize the surface. The best parameters for 3D printing with the polymyrcene resins were an exposure of 10 s at an intensity of 10 mW/cm^2^ per 50 µm slice to produce template molds. This study demonstrated the possibility of using a bio-based material for 3D printing, and the versatility and width of different tunable properties.

Shimpf et al. [108] tailored limonene-based dimethacrylate for 3D applications. They synthesized limonene dimethacrylate (LDMA) resins, starting from limonene oxide. Different oligomers were prepared, and different limonene oxide and methacrylic acid ratios were tested. The tested resins were a blend of a base formulation (commercial product) and the same formulation, but with the addition of the limonene-based resin. The addition involved up to 50 wt %. Some 3D-printed objects are presented in Figure 21. The advantage of using LDMA lies in the low viscosity and in the possibility of tailoring the properties by mixing the resin with a commercial one. Moreover, the high stiffness and strength persisted after the blending, and the addition of LDMA even enabled the Tg to be improved.

In 2020, Ortiz et al. [109] selected a terpenoid-like nopol as the starting material to prepare epoxy monomers. Nopol is a terpene derivate that was bi- and tri-epoxidized. Three different protocols were followed: alkylation of the nopol with epichlorohydrin; aromatic substitution of the chlorine atom in the cyanuric chloride with the alkoxide of nopol; and preparation of an acetal by means of the reaction between nopol and 3-cyclohexene 1-carboxyaldehyde under an acid catalyst. Iodonium salt was used to start the photopolymerization under UV. The resulting networks had Tg over a 61 to 66 °C range.

Li et al. proposed one of the latest studies available on terpenes [110]. They used limonene in a powder coating. A thiol-ene network was formed by reacting trimethylolpropane tris(3-mercaptopropionate) with poly(limonene carbonate)s (PLCs). The properties of the UV-cured powder coating showed great potential for this renewable starting material. The coating showed high transparency, good acetone resistance, high pencil hardness and high König hardness.

## 3. Conclusions

In this review, we have reported the exploitation of bio-based monomers, derived from industrial pulp and paper processing, with UV-curing technology. The byproducts of industrial processes, such as those of the pulp and paper industry, can become suitable starting points to make valuable chemicals. Bio-derived raw materials, such as lignin, can be used as they are or can be modified to achieve photocurable monomers, like vanillin. Vanillin as an interesting example of a chemical derived from a green source that is industrially available and has great potential for functionalization, which can lead to its use as a monomer for UV-curing, is reported here. Rosin and terpenes are another two examples of families that are readily available from the pulp and paper industry, which can be used for UV-curing. The chemical structure of these families offers a good structural property base that can be an important starting point to synthetize competitive bio-based resins. Thus, the final properties of the new materials may be comparable with petroleum-based materials. Of all the considered products, limonene appears to be particularly interesting, due to its easy availability and intrinsic properties. Several of the reported studies have in fact demonstrated its potential for UV-curing.

Hence, the aforementioned resins may be used for coating applications using thiol-ene chemistry, epoxy chemistry or acrylate chemistry. In the latter case, the functionalization can modify the different biomolecules, thereby allowing them to be used in 3D printing. Several examples that show the great potential of green-based materials for this new advanced manufacturing process, have been presented in this review.

However, the low usage of the by-product for chemical is a limitation in terms of availability and development of new processes and monomers. The main fraction is devoted to energy (98%).The future challenge will be to ensure that specific processes isolate the starting monomers and oligomers in high yield and purity. These aspects may contribute to obtain competitive commodities and products with respect to the petroleum-based, allowing a greener future. The environmental cost of the entire processes should be considered and evaluated to find the best opportunities for the use of pulp and paper by-products.

## Figures and Tables

**Figure 1 polymers-13-01530-f001:**
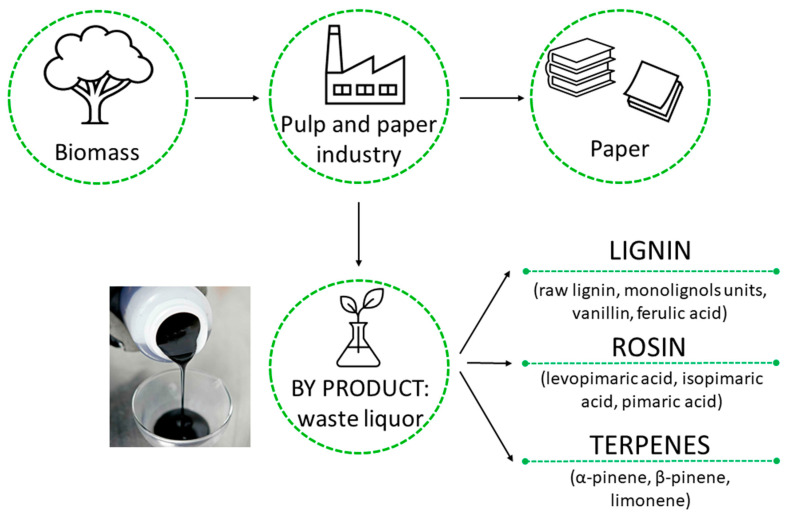
The main families of constituents present in the by-products derived from pulp and paper industry. Lignin, rosin and terpenes used as a platform for valuable chemicals.

**Figure 2 polymers-13-01530-f002:**
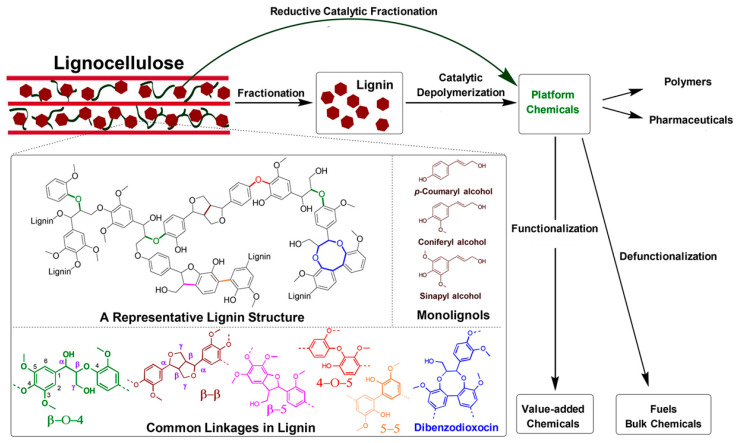
The main chemical bonds present in the chemical structure of lignin. Adapted with permission from [8]. *Copyright © 2021, American Chemical Society*.

**Figure 3 polymers-13-01530-f003:**
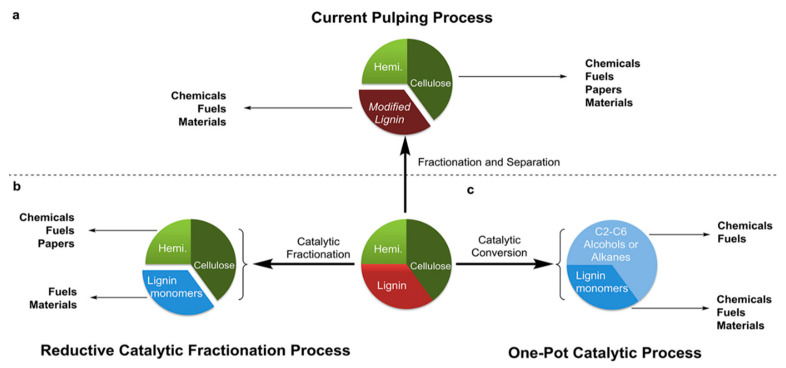
Catalytic processes for lignin depolymerization to obtain new platform chemicals. Comparison between the different starting materials used. (**a**) Lignocellulose fractionation prior to catalytic processing; (**b**) Reductive catalytic fractionation (RCF) in the presence of a catalyst; (**c**) complete conversion of all lignocellulose mass by one-pot catalytic processing. Adapted with permission from [8]. *Copyright © 2021, American Chemical Society*.

**Figure 4 polymers-13-01530-f004:**
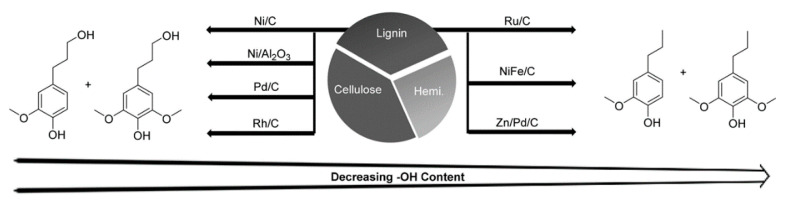
Chemical structures of the main different products obtained from different heterogenous catalytic systems. Adapted with permission from [8], *Copyright © 2021, American Chemical Society*.

**Figure 5 polymers-13-01530-f005:**
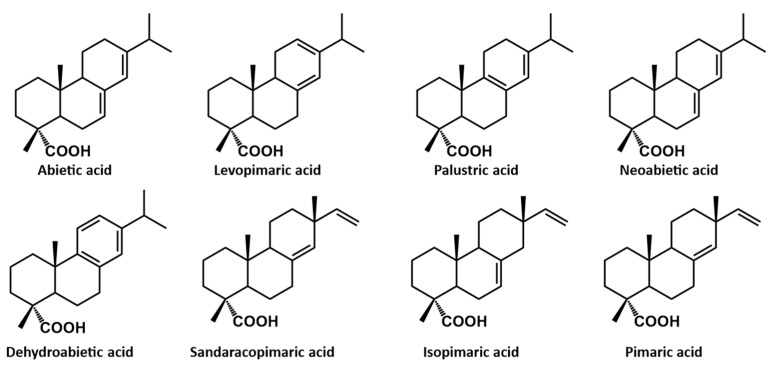
The chemical structure of rosin derivates [38]. (Open access source: doi:10.3390/molecules24091651).

**Figure 6 polymers-13-01530-f006:**
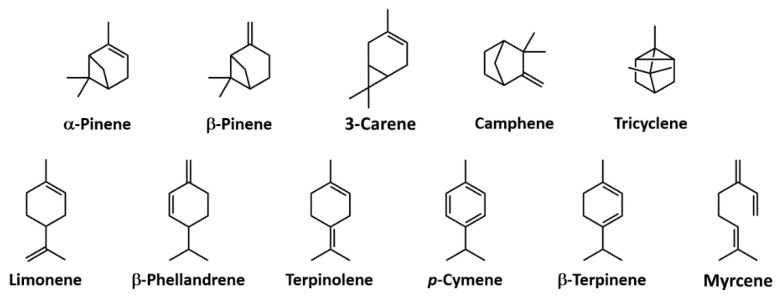
Chemical structures of terpenes in turpentine. Reprinted with permission from [41], *Copyright © 2021 Elsevier*.

**Figure 7 polymers-13-01530-f007:**
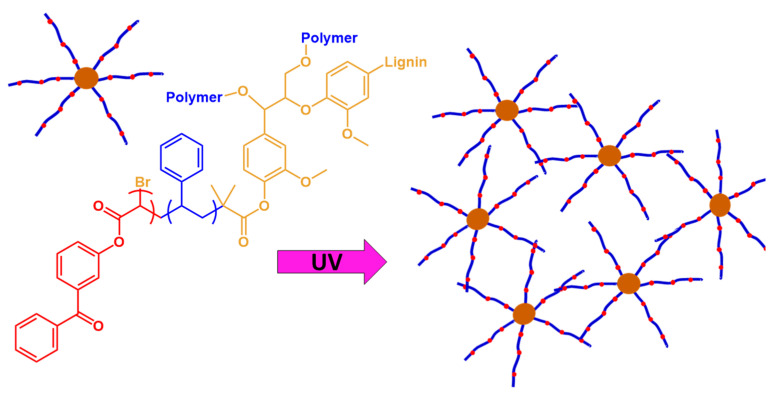
Schematic illustration of UV-curable thermoplastic lignin-grafted copolymers. Adapted with permission from [57]. *Copyright © 2021, American Chemical Society*.

**Figure 8 polymers-13-01530-f008:**
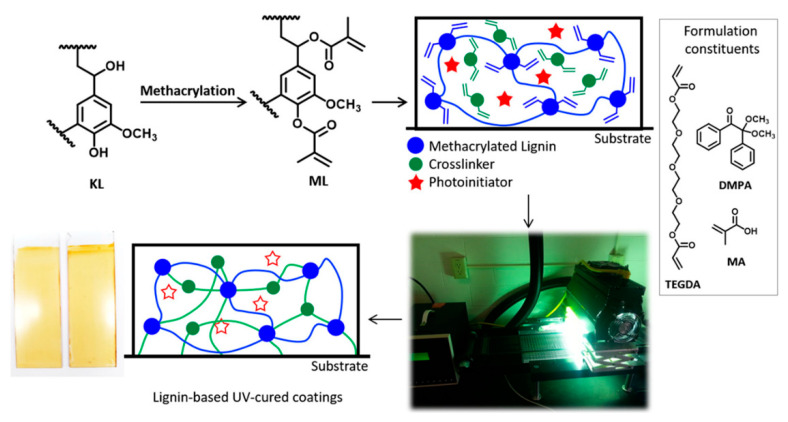
Illustration of the synthesis of ML and the curing process. Adapted with permission from [59]. *Copyright © 2021, American Chemical Society*.

**Figure 9 polymers-13-01530-f009:**
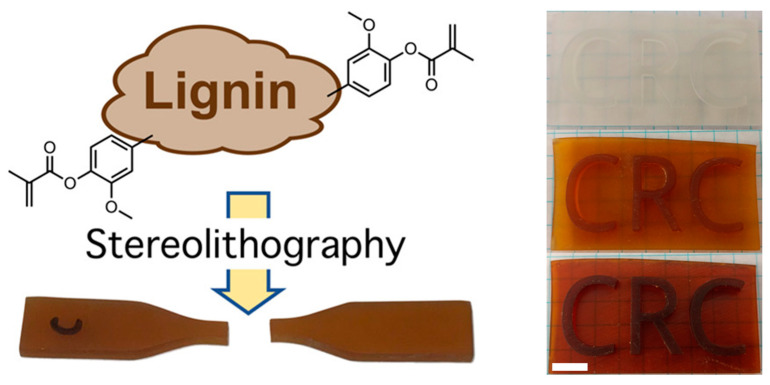
Printed material, from the top: 0 wt %, 5 wt % and 10 wt % of lignin resin. Adapted with permission from [62]. *Copyright © 2021, American Chemical Society*.

**Figure 10 polymers-13-01530-f010:**
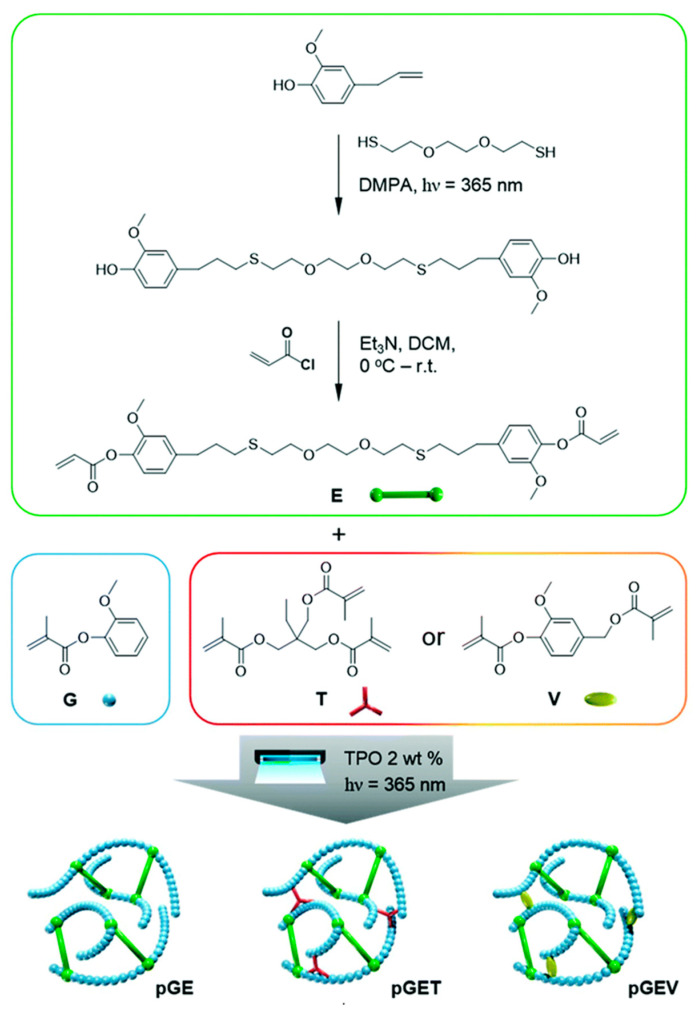
Schematic route followed to achieve the different tested formulations. Reproduced from Reference [63] with permission from the *Royal Society of Chemistry*.

**Figure 11 polymers-13-01530-f011:**
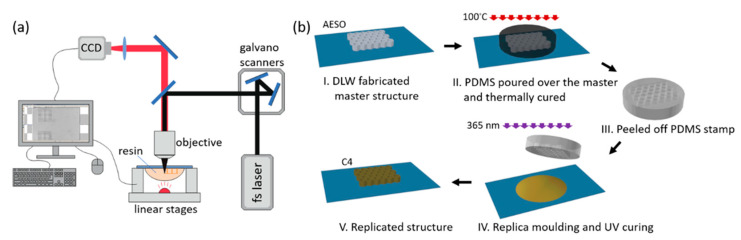
A schematic view of the two 3D printing techniques used; (**a**) direct laser writing (DWL) and (**b**) microtransfer molding technique μTM [66]. (Open access source: doi:10.3390/polym12020397).

**Figure 12 polymers-13-01530-f012:**
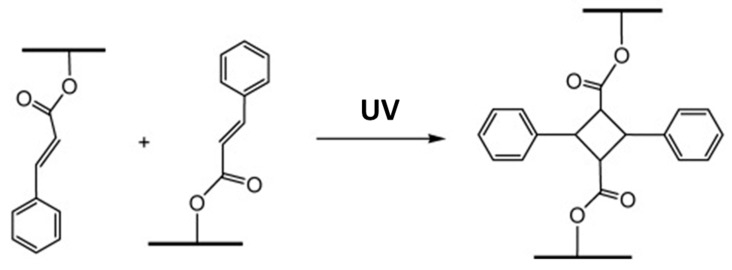
Photo-crosslinking between cinnamate moieties. Reprinted with permission from [78], *Copyright © 2021 Elsevier*.

**Figure 13 polymers-13-01530-f013:**
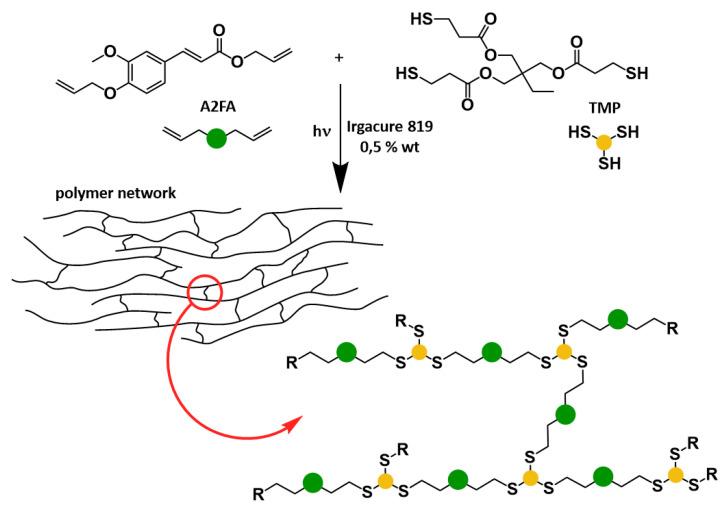
A schematic illustration showing the thiol-ene photopolymerization of h-A2FA and TRIS in the presence of Irgacure 819. Reprinted with permission from [83], *Copyright © 2021 Elsevier*.

**Figure 14 polymers-13-01530-f014:**
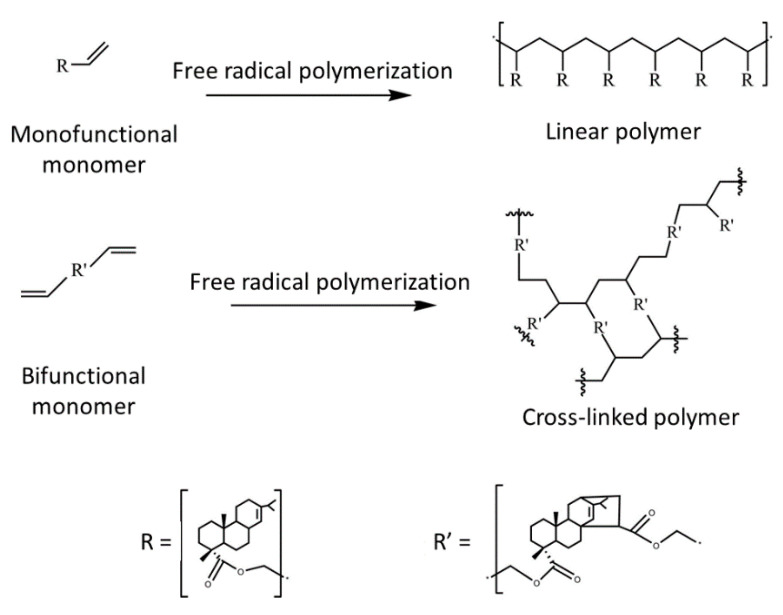
Proposed structures for the cured film of mono and bifunctional monomers in the work by Lu et al. Reprinted with permission from [91], *Copyright © 2021 Elsevier*.

**Figure 15 polymers-13-01530-f015:**
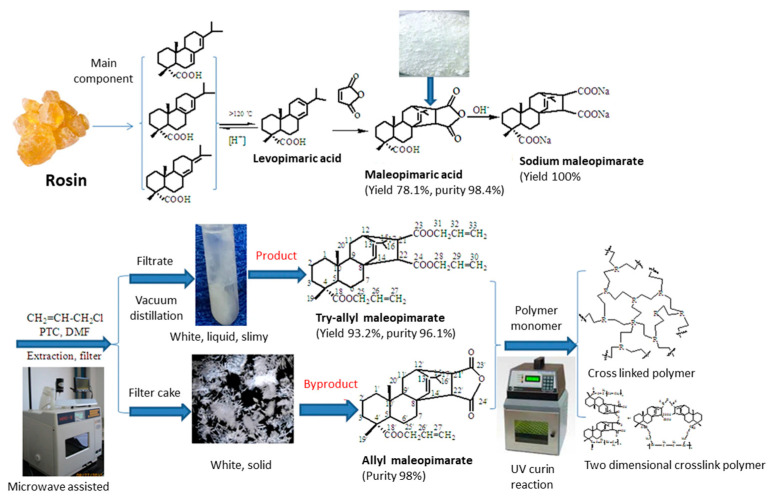
From rosin to the final coating as proposed by Lu et al. [92]. (Open access: doi:10.1038/s41598-018-20695-5).

**Figure 16 polymers-13-01530-f016:**
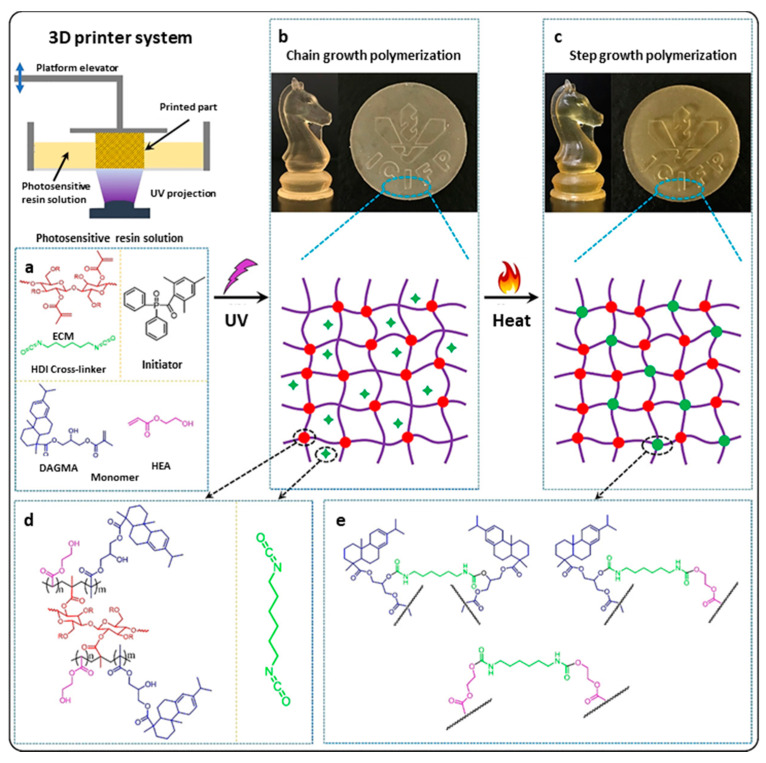
Scheme of structures formed in 3D printing by a photocuring process and by a dual process (UV + thermal curing). Adapted with permission from [94], *Copyright © 2021 Elsevier*.

**Figure 17 polymers-13-01530-f017:**
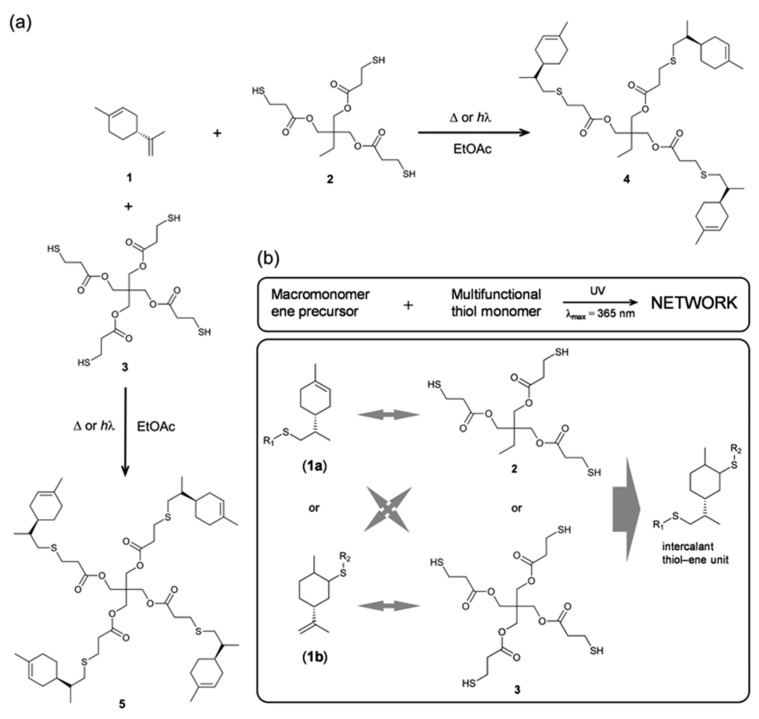
(**a**) Formation of prepolymer (**4** and **5**) induced thermally or photochemically; (**b**) UV-curable network formation by thiol-ene chemistry. Possible arrangement of pendant residue (**1a** and **1b**) [103], published by *The Royal Society of Chemistry*.

**Figure 18 polymers-13-01530-f018:**
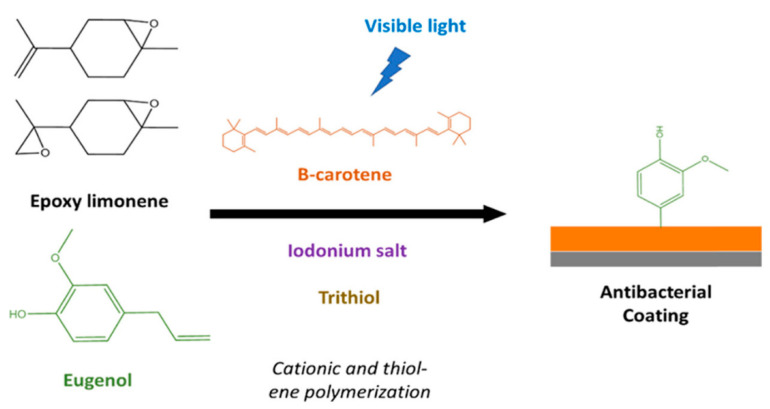
Scheme of coating formation triggered by visible light. Adapted with permission from [104]. *Copyright © 2021, American Chemical Society*.

**Figure 19 polymers-13-01530-f019:**
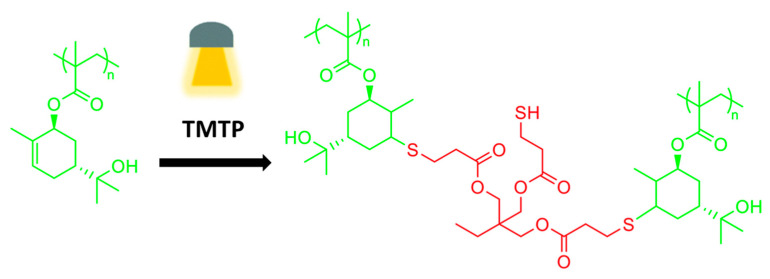
UV-crosslinking of methacrylated polysobrerolmethacrylate [105], published by *The Royal Society of Chemistry*.

**Figure 20 polymers-13-01530-f020:**
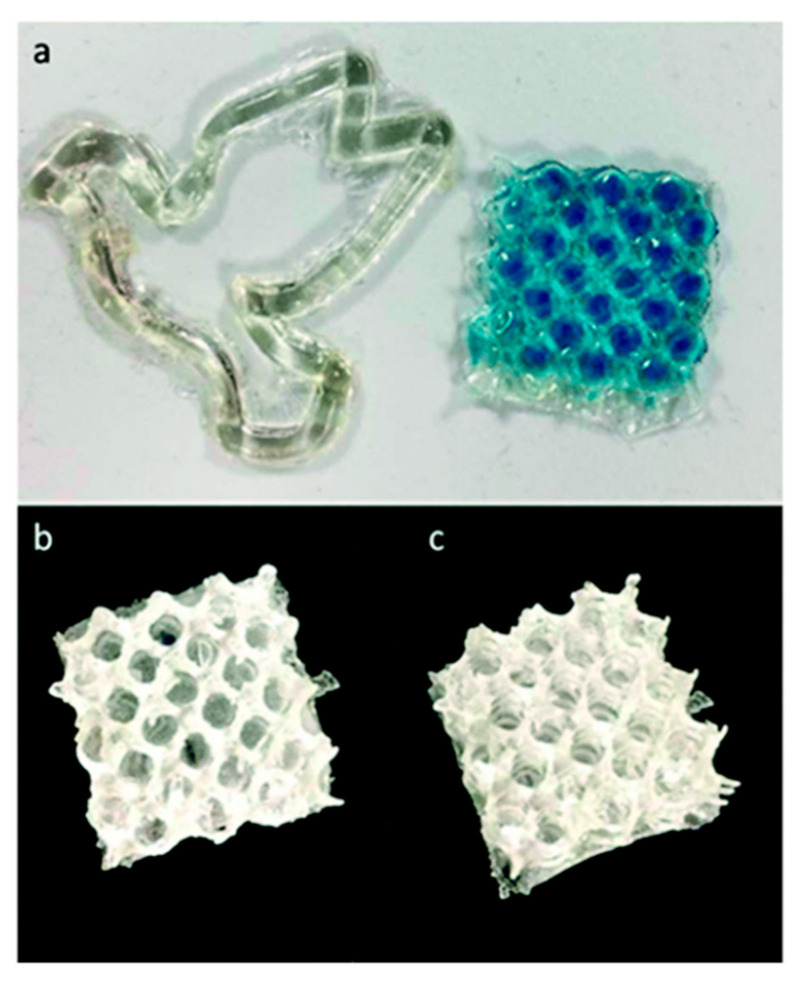
Examples of 3D-printed structures from (**a**,**b**) linalool and (**c**) limonene prepolymer resins [106], published by *The Royal Society of Chemistry*.

**Figure 21 polymers-13-01530-f021:**
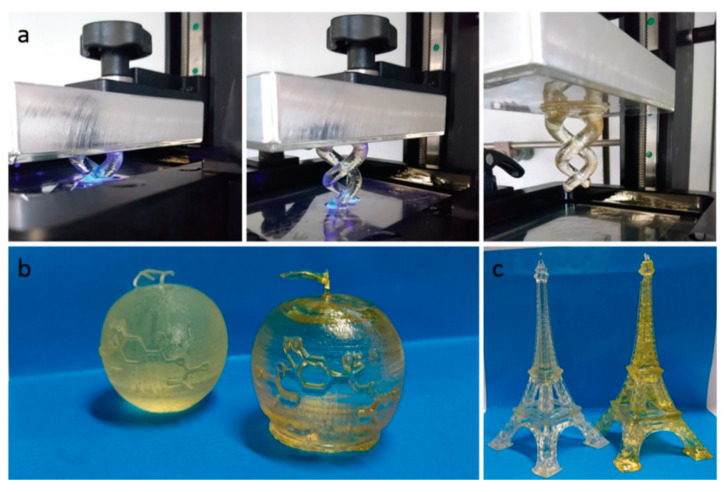
Examples of 3D printed objects: (**a**) double helix; (**b**) orange model; (**c**) model of Eiffel Tower; base resin was used as control (object on the left in pictures (**b**,**c**), printed formulation containing LDMA on the right of the cited imagines [108]. (Open access source: doi:10.1002/mame.202000210).

**Table 1 polymers-13-01530-t001:** The different percentages of cellulose, hemicellulose, and lignin in different plants. Reprinted with permission from [3], *Copyright © 2021 Elsevier*.

Lignocellulosic Materials	Cellulose (%)	Hemicellulose (%)	Lignin (%)
Hardwood stems	40–55	24–40	18–25
Softwood stems	45–50	25–35	25–35
Nut shells	25–30	25–30	30–40
Corn cobs	45	35	15
Wheat straw	30	50	15
Rice straw	32	24	18
Leaves	15–20	80–85	0
Grasses	25–40	25-50	10–30
Switch grass	31–32	35–50	20–25
Sugarcane bagasse	42	25	20
Sweet sorghum	45	27	21
Cotton seed hairs	80–95	5–20	0
Coconut husk	39	16	30
Sorted refuse	60	20	20
Paper	85–99	0	0–15
Newspaper	40–55	25–40	18–30
Waste paper from chemical pulps	60–70	10–20	5–10
Primary wastewater solids	8–15	NA	24–29

**Table 2 polymers-13-01530-t002:** The percentages of the three main alcohols in the principal plant families. Reprinted with permission from [20], *Copyright © 2021 Elsevier*.

Monolignol	Broadleaf Wood (%)	Conifer Wood (%)	Grass (%)
Sinapyl alcohol (S)	50–75	0–1	25–50
Coniferyl alcohol (G)	25–50	90–95	25–50
p-Coumaryl alcohol (H)	Trace	0.5–3.4	10–25

**Table 3 polymers-13-01530-t003:** Phenolic acids derived from lignin. Reprinted with permission from [20], *Copyright © 2021 Elsevier*.

Phenolic Acid and Aldehydes	Lignin Fraction (% yield, *w*/*w*)				
	OPF Lignin	Caligonum Monogoliacum Lignin	Tamarix spp. Lignin	Maize Stem Lignin	Rice Straw Lignin
*p*-Hydroxybenzoic acid	0.42	1.68	1.67	0.82	1.12
*p*-Hydroxybenzaldehyde	0.49	1.35	1.21	2.48	1.59
Vanillic acid	0.25	1.04	1.14	0.03	0.36
Syringic acid	0.84	2.16	1.77	1.28	1.82
Vanillin	1.02	17.96	18.12	10.49	15.49
Syringaldehyde	2.60	9.36	10.34	13.05	13.00
*p*-Coumaric acid	N/A	3.08	2.55	0.32	0.61
Ferulic acid	0.30	0.91	0.47	0.82	1.22
Molar ratio (S:G:H)	58:22:15	58:22:15	31:59:10	N/A	N/A

**Table 4 polymers-13-01530-t004:** Summary of the use of lignin in the aforementioned studies.

Lignin Type	Maximum Amount (wt %)	Role of Lignin	Reference
Acell lignin	20	filler	Rozman [53]
Kraft lignin	1	filler	Zhang [54]
Organosolv lignin	3	filler	Ibrahim [55]
Alkali lignin (hydroxymethylated)	12	monomer	Chao [56]
Kraft lignin (copolymerized)	-	monomer	Wang [57]
Kraft lignin (methacrylated)	30	monomer	Hajirahimkhan [58]
Kraft lignin (methacrylated)	31	monomer	Hajirahimkhan [59]
Organosolv lignin (epoxy acrylate)	25	monomer	Yan [60]
Organosolv lignin (epoxy acrylate)	25	monomer	Yan [61]
Organosolv lignin (acrylate)	15	monomer	Sutton [62]

## Data Availability

Not applicable.

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
