# Peer review of "UV-Curable Bio-Based Polymers Derived from Industrial Pulp and Paper Processes"

_polymers, 2021, doi:10.3390/polym13091530_

Round 1

Reviewer 1 Report

The  article title cover a good field of research, However as a review, the following notes need to consider

  1. in the title " UV-Curable Bio-based monomers derived from industrial pulp and paper processes" authors mentioned UV-curing of monomers, while main parts of article especially in lignin part they  just covered UV curing of lignin which is a macromolecule not monomer. 
  2. Big part of the review is about lignin catalytic degradation which is out of this review scope and already covered by other reviews comprehensively. Also there is almost no review about lignin monomer UV curing in the article.   
  3. There is no perspective in the review about the different UV curing methods, mechanism of reaction and results.                    

Author Response

the response to the reviewer comments are reported in the attached file

Reviewer 2 Report

I think this is an excellent review, and should be published in polymers

Author Response

We acknowledge the very positive comments of this reviewer. 

Reviewer 3 Report

This paper presentes the recent advances in the application of pulp derived monomers and oligomers in the UV-initiated formation of linear and crosslinked polymers. This review is interesting for the wide community of researchers in the fields of polymer and green chemistry. It is applicable for publication in Polymers. However for benefit of the readers it will be good if authors will take into account the following remarks:

  1. It is rather not easy to understand the origin and the “tree” of the products derived from the pulp during reading section 1. It seems that it will be better to make some introduction generalizing the text in sections 1.1, 1.2 and 1.3. Also some scheme demonstrating the hierarchy of pulp based chemicals is strongly needed. Additionally, Figure 2 is very difficult to understand.
  2. Some future perspectives and problems, which are need to be solved should be more emphasized in the conclusion.
  3. The sequence of references on the figures should be checked and corrected trough the manuscript. For example in text it is written “ Wang et al. [53] prepared UV curable lignin thermoplastic copolymers, as shown in Figure 7”, while it seems that Figure 6 should be referenced here. The same problems seems to occur with Figures 7, 8 and 9.

Author Response

The answer to the reviewer is reported in the attached file

Round 2

Reviewer 1 Report

The review title is not matched with content. more than 95% on pulping black liquor are lignin. Authors filled big part of the article with lignin degradation method into small molecules to show importance of small molecule from lignin, while there is almost no review about curing of small compounds derived from lignin in the review. 

It is better to change review article into " UV-curing of polymers derived from pulping process" and removing lignin degradation part from the article. 

Author Response

The answers to the reviewer are reported in the attached file
